# Outcomes of minimal change disease without nephrotic range proteinuria

**Hyung Eun Son**[1,2], **Giae Yun**[3], **Eun-Jeong Kwon**[3], **Seokwoo Park**[3,4], **Jong Cheol Jeong**[3], **Sejoong Kim**[3,5], **Ki Young Na**[3,5], **Jin Ho Paik**[6,7], **Ho Jun Chin**[3,5]*

1 Department of Internal Medicine, Chung-Ang University Gwangmyeong Hospital, Gwangmyeong, Korea, 2 Department of Internal Medicine, Chung-Ang University College of Medicine, Seoul, Korea, 3 Department of Internal Medicine, Seoul National University Bundang Hospital, Seong-nam, Korea, 4 Department of Biomedical Sciences, Seoul National University College of Medicine, Seoul, Korea, 5 Department of Internal Medicine, Seoul National University College of Medicine, Seoul, Korea, 6 Department of Pathology, Seoul National University Bundang Hospital, Seong-nam, Korea, 7 Department of Pathology, Seoul National University College of Medicine, Seoul, Korea

* mednep@hanmail.net

**Data Availability Statement:** All relevant data are within the paper and its Supporting Information files.

**Funding:** The authors received no specific funding for this work.

## Abstract

Minimal change disease (MCD) is characterized by edema and nephrotic range proteinuria (NS). However, the fate of MCD without nephrotic proteinuria requires elucidation. We retrospectively reviewed 79 adults diagnosed with primary MCD at their initial renal biopsy at a tertiary hospital between May 2003 and June 2017. Clinicopathologic features were compared between patients with and without NS. The frequency of flaring to nephrotic proteinuria and renal outcomes were assessed during follow-up. There were 20 and 59 patients in the Non-NS and NS groups, respectively. The Non-NS group had a lower frequency of acute kidney injury (AKI) during the follow-up period [5.0% vs. 59.3%, p <0.001]. The response rate to steroid treatment was 100% in the Non-NS group and 92.3% in the NS group (p = 1.000). Except for one patient, the Non-NS group was treated with steroids when their proteinuria increased to a nephrotic level. There were no differences in the frequency of the first relapse or the number of relapses among patients with initial remission from nephrotic range proteinuria. At the final visit, the complete remission rate was 73.4%. The estimated glomerular filtration rate during follow-up was significantly better in the NS group than the Non-NS group, given the higher rates of AKI at renal biopsy. The rates of renal events, end-stage renal disease, and mortality did not differ between the groups. Adult MCD patients with nephrotic and non-nephrotic range proteinuria showed similar outcomes. Accordingly, this population must be carefully managed, regardless of the amount of proteinuria at renal biopsy.

## Introduction

The primary nephrotic syndrome presents with heavy proteinuria, hypoalbuminemia, hyperlipidemia, and clinical symptoms such as edema or pleural effusion. Chronic glomerulonephritis is reportedly the third highest cause of progression to end-stage renal disease (ESRD) in

**Competing interests:** The authors have declared that no competing interests exist.

South Korea [1]. Minimal change disease (MCD) comprises approximately 11–20% of adult primary nephrotic syndrome cases [2]. Among 21,426 adults that underwent a kidney biopsy in South Korea, 9.17% were diagnosed with MCD [3], making it the third most prevalent primary glomerulonephritis following IgA nephropathy and membranous nephropathy [4]. MCD often presents with abrupt onset of edema and full nephrotic syndrome. Hypertension, microhematuria, and acute renal failure may also develop. Compared with children, a smaller percentage of adults with MCD show features of nephrotic syndrome. Diagnosis of MCD in adults is based on histopathological characteristics, and adequate samples that include more than 10 glomeruli are needed for a 95% chance of detection [5].

In adults, MCD exhibits a relatively large level of proteinuria compared with other primary glomerulonephritis. Until this study, most studies have focused on heavy proteinuria diagnosed with MCD. Therefore, the long-term prognosis of adult MCD without nephrotic syndrome as an initial manifestation is unclear. To fully understand MCD, the prognosis of MCD with typical pathologic findings needs to be defined in terms of clinical characteristics, response rates, and relapse rates in patients with and without nephrotic syndrome. In this study, we compared the clinicopathologic features and outcomes between MCD patients with and without heavy proteinuria for approximately 5 years.

## Materials and methods

### Patients

We retrospectively reviewed 1,516 adult candidates aged ≥ 18 years who underwent renal biopsy between May 2003 and June 2017 at a single center in South Korea: Seoul National University Bundang Hospital. Renal biopsy was performed using the ultrasonography-guided percutaneous gun biopsy technique. We collected the pathological findings on light, immunofluorescence, and electron microscopy (EM), and all biopsy specimens were initially evaluated by an independent renal pathologist blinded to patients' outcomes. The detailed histology reports and biopsy slides of patients with MCD were re-reviewed. We defined MCD pathologically as near-normal findings on light microscopy, except for the mild expansion of the mesangium or global glomerulosclerosis, which could also be seen as nonspecific findings. Segmental sclerosis was not seen. There were no complement or immunoglobulin deposits on immunofluorescence microscopy. Diffuse effacement of podocyte foot processes was seen on EM. We excluded patients with biopsy samples that included fewer than 10 glomeruli; patients with possible secondary causes, such as malignancy or lupus nephritis; patients who had immunosuppressive treatment before renal biopsy; patients with fewer than three proteinuria tests after the renal biopsy; and patients followed for < 3 months after the renal biopsy (Fig 1). In total, 79 patients were included in this study.

### Data collection

The patient's clinical characteristics, laboratory results, renal pathology results, and medication prescriptions during each follow-up period were collected from their electronic health records (EHRs) from the period of renal biopsy to the final follow-up visit, with the primary query being the patients' identification number. The baseline data collected during the biopsy included age, sex, systolic and diastolic blood pressures, comorbidities, and medications. Comorbid conditions included a history of hypertension, diabetes, coronary heart disease, and cerebrovascular disease. Hypertension was defined as systolic blood pressure of ≥140 mmHg, diastolic blood pressure of ≥90 mmHg, or taking anti-hypertensive medications. We collected laboratory data, including serum cholesterol levels, glucose, total protein, albumin, hemoglobin, isotope dilution mass spectrometry (IDMS)-traceable creatinine, and spot urine protein

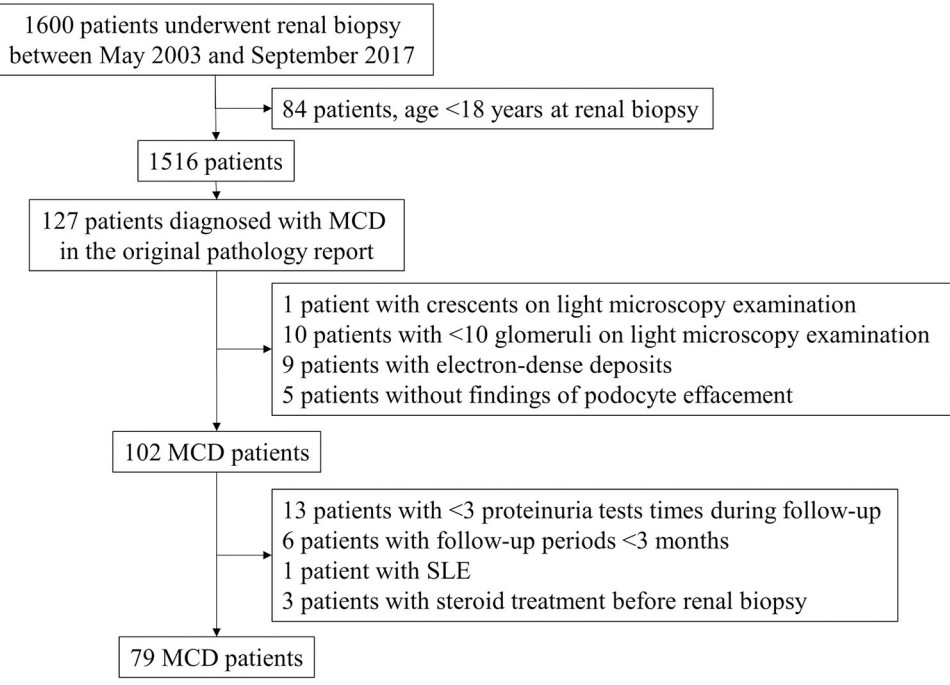

**Fig 1. Selection of patients.** MCD: Minimal change disease, SLE: Systemic lupus erythematosus.

to creatinine ratio (UPCR). The estimated glomerular filtration rate (eGFR) was calculated using the Chronic Kidney Disease Epidemiology Collaboration (CKD-EPI) equation [6]. Acute kidney injury (AKI) was defined according to the Kidney Disease Improving Global Outcomes (KDIGO) criteria [7]. Specifically, AKI was diagnosed based on the change in serum creatinine concentration from the lowest value during the follow-up period to the value measured on the biopsy day. The stages of AKI were as follows stage 1, $\geq$0.3 mg/dL absolute or 1.5- to 2.0-fold relative increase in serum creatinine; stage 2, > 2.0- to 3.0-fold increase in serum creatinine; and stage 3, >3.0-fold increase in serum creatinine or serum creatinine $\geq$4.0 mg /dL with an acute rise of >0.5 mg/dL. Nephrotic range proteinuria was defined as over 3.0 g/g in a spot urine sample. Complete remission (CR) of proteinuria, relapse of proteinuria, and steroid dependency was defined according to the revised KDIGO guideline for glomerulo-nephritis [8] (Table 1). Microscopic hematuria was defined as $\geq$5 erythrocytes/high power field on urine sediment microscopy. ESRD events were collected from the ESRD registry of the Korean Society of Nephrology, and mortality events from our hospital's database or the

**Table 1. Definitions of MCD courses.**

| MCD courses | Definition |
| --- | --- |
| **Nephrotic range proteinuria** | UPCR $\geq$3.0 g/g creatinine |
| **Complete remission of proteinuria** | UPCR <0.3 g/g creatinine |
| **Relapse of proteinuria** | The reappearance of nephrotic range proteinuria after remission |
| **Steroid dependency** | Relapse of proteinuria during steroid tapering or within 2 weeks after cessation of steroids |

MCD: Minimal change disease, UPCR: Urine protein/creatinine ratio.

Ministry of the Interior and Safety Korea database after merging with our data using patients' unique identification numbers.

### Kidney pathology

As previously described [9], pathologic diagnosis contains the following contents. Glomerular lesions such as global sclerosis, segmental sclerosis, glomerular ischemic change, and crescentic changes were reported as a proportion of the total glomeruli in an evaluated specimen. A semi-quantitative assessment of changes in mesangial cellularity, mesangial matrix, tubular atrophy, and interstitial inflammation and fibrosis was performed, with the results classified as normal, mild, moderate, moderate to severe, and severe. Vascular abnormality was defined as the presence of arteriolar hyalinosis and arteriosclerosis. The immunofluorescence study used the classic direct technique with antibodies against eight proteins (IgG, IgM, IgA, C3, C1q, fibrinogen, kappa, and lambda chains). The results were reported semi-quantitatively as negative (0), trace (0.5), and 1–3 positive (1–3). After EM analysis was done in all biopsy samples, findings on EM were described as follows; the presence of electron-dense deposits in the area of the mesangium, subendothelium, and subepithelium; and severity of foot process effacement of the podocytes, which were reported as none, focal (mild, moderate, moderate to severe, severe), or diffuse.

### Outcomes

Renal events were defined as any decrease in eGFR by more than 50% during a follow-up visit compared with that at renal biopsy, eGFR $<15$ ml/min/1.73 m$^2$, or development of ESRD during the follow-up period. The slope of GFR change per year (ml/min/1.73 m$^2$/year) was defined as the difference between eGFR measured on the day of the biopsy and the last measurement, divided by the number of years of follow-up. We compared the achievement of CR, the number of relapses of proteinuria after remission, the slope of eGFR change during the follow-up period, steroid dependency rates, and the development of renal events such as AKI, ESRD, and mortality between the groups.

### Exposure

Patients were divided into two groups according to the UPCR measured on the day of the renal biopsy. Patients with UPCR $\geq 3.0$ g/g were classified as the nephrotic group (NS group), and those with UPCR $<3.0$ g/g were classified as the non-nephrotic group (non-NS group).

### Statistical analysis

Data are presented as mean ± standard deviation (SD) for continuous variables and number (percentages) for categorical variables. Continuous variables were compared using the Mann-Whitney U test, and categorical variables using Pearson's Chi-square test or Fisher's Exact test according to the number of cells. The relationship between variables was assessed by linear and logistic regression tests for continuous and dichotomized variables. Kaplan-Meier analysis was used to evaluate survival curves for CR, relapse of proteinuria, renal events, ESRD, and mortality. Risk factors for CR, relapse of proteinuria, renal events, ESRD, or mortality were identified using adjusted Cox proportional models. Statistical significance was set at a $p<0.05$. All the analyses were performed using SPSS Statistics for Macintosh, Version 22 (IBM, Armonk, NY) and Stata software, Version 17 (Stata Corp LLC., College Station, TX).

### Ethical approval

This study was approved by the institutional review board (IRB) of Seoul National University Bundang Hospital (IRB approval No. B-1910-572-304). The IRB waived written consent because of the study's retrospective nature. All data were anonymized entirely before access to the database.

## Results

### Baseline characteristics

Among the 79 patients included, the mean age at renal biopsy was 53.7 ± 19.2 (range: 18.5–99.0) years, and there were 38 men (48.1%). The highest level of UPCR within the 6 months before the renal biopsy was 9.04 ± 6.34 (range: 0.38–36.66) g/g creatinine. There were 16 patients (20.3%) with a pre-biopsy UPCR <3.00 g/g creatinine, among whom two exhibited UPCRs <0.30 g/g creatinine at admission for renal biopsy. Another patient with nephrotic range UPCR before renal biopsy (1/63) also had a UPCR <0.30 g/g creatinine at admission. This suggests that spontaneous remission of proteinuria without immunosuppressive therapy occurred in three patients during the 6 months wait for renal biopsy.

The patients were divided into Non-NS (n = 20, 25.3%) and NS-groups (n = 59, 74.7%) according to the presence of proteinuria $\geq$ 3g/g on the day of the biopsy. The groups' demographics and clinic-pathologic findings are summarized in Table 2. The groups had no differences in demographic features, underlying medical illnesses, and blood pressures at renal biopsy. However, there was a difference in the degree of proteinuria. UPCR within 6 months before renal biopsy was 11.22 ± 6.25 g/g creatinine in the NS group and 2.63 ± 1.92 g/g creatinine in the Non-NS group (p <0.001). The UPCR at renal biopsy was also higher in the NS group (10.16 ± 6.21 vs. 1.36 ± 1.00 g/g creatinine, p <0.001). The eGFR was lower (75 ± 36 vs. 98 ± 27 ml/min/1.73 m$^2$, p = 0.012), and the prevalence of AKI at admission for renal biopsy was higher in the NS group (59.3% vs. 5.0%, p <0.001). Across both groups, there were 14 patients with stage 1 AKI, nine with stage 2, and 13 with stage 3 at admission for renal biopsy. Dialysis was only needed in five patients (8.3%) in the NS group. Serum protein and albumin levels were lower in the NS group when the total cholesterol level was higher (p <0.001, Table 2). Four out of 20 patients in the Non-NS group had a max UPCR $\geq$ 3g/g creatinine before the biopsy. When dividing patients according to the maximal amount of proteinuria 6 months before biopsy by 3g/g creatinine, 16 patients had < 3 g/g creatinine of proteinuria, and the other 63 patients had > 3g/g creatinine of proteinuria. When comparing groups according to a max UPCR before biopsy, the two groups showed a high UPCR, lower GFR, and a higher proportion of AKI (S1 Table).

The renal specimens obtained for pathological examination included enough glomeruli for MCD diagnosis (10–86 glomeruli). Notably, most pathologic findings were similar between the groups, except for interstitial changes and podocyte effacement. The presence of interstitial fibrosis and inflammation was more prevalent in the NS group (p <0.05, S2 Table), as was diffuse effacement of the podocytes under EM examination (93.2% vs. 40.0%, p <0.001). The severity of podocyte effacement was positively correlated to the amount of proteinuria and the presence of nephrotic range proteinuria according to the adjusted linear and logistic regression models, respectively (S3 Table).

### Clinical course

During follow-up after renal biopsy (mean: 62.9 ± 50.8 months, range: 3.0–201.7 months), a UPCR test was conducted 28.5 ± 19.0 (range: 3–142) times; at a mean of 0.64 ± 0.49 times/

**Table 2. Characteristics of patients with minimal change disease according to the amount of proteinuria.**

| Characteristic | Completeness of data (%) | Non-NS (n = 20) | NS (n = 59) | p-value |
|---|---|---|---|---|
| Age (years) | 100.0 | 53.0 ± 15.2 | 53.9 ± 20.5 | 0.550 |
| Male (n, %) | 100.0 | 8 (40.0) | 30 (50.8) | 0.447 |
| DM (n, %) | 100.0 | 5 (25.0) | 7 (11.9) | 0.168 |
| Hypertension (n, %) | 100.0 | 10 (50.0) | 32 (54.2) | 0.799 |
| History of CHD (n, %) | 100.0 | 0 (0.0%) | 4 (6.8%) | 0.567 |
| Weight (kg) | 100.0 | 61.8 ± 11.4 | 66.2 ± 12.8 | 0.169 |
| SBP (mmHg) | 100.0 | 118 ± 15 | 125 ± 18 | 0.118 |
| DBP (mmHg) | 100.0 | 69 ± 11 | 73 ± 11 | 0.181 |
| Cholesterol (mg/dl) | 93.7 | 262 ± 144 | 372 ± 116 | <0.001 |
| Glucose (mg/dl) | 100.0 | 127 ± 33 | 108 ± 29 | 0.012 |
| Protein (g/dl) | 92.4 | 5.7 ± 1.2 | 4.3 ± 0.7 | <0.001 |
| Albumin (g/dl) | 92.4 | 3.3 ± 1.0 | 2.1 ± 0.5 | <0.001 |
| Hemoglobin (g/dl) | 100.0 | 13.4 ± 1.8 | 13.5 ± 2.1 | 0.664 |
| Creatinine (mg/dl) | 100.0 | 0.89 ± 0.81 | 1.23 ± 0.81 | 0.008 |
| Creatinine before biopsy (mg/dl) | 100.0 | 0.65 ± 0.21 | 0.66 ± 0.24 | 0.871 |
| GFR (ml/min/1.73 m2) | 100.0 | 98 ± 27 | 75 ± 36 | 0.012 |
| AKI at biopsy (n, %) | 100.0 | 1 (5.0) | 35 (59.3) | <0.001 |
| Stage 1 (n, %) | 100.0 | 0 (0.0) | 14 (23.7) | <0.001 |
| Stage 2 (n, %) | 100.0 | 0 (0.0) | 9 (15.3) | |
| Stage 3 (n, %) | 100.0 | 1 (5.0) | 12 (20.3) | |
| AKI requiring dialysis (n, %) | 100.0 | 0 (0.0) | 5 (8.3) | 0.322 |
| Max. UPCR before biopsy (g/g cr) | 100.0 | 2.63 ± 1.92 | 11.22 ± 6.25 | <0.001 |
| <0.30 g/g cr (n, %) | 100.0 | 0 (0.0) | 0 (0.0) | <0.001 |
| 0.30–2.99 g/g cr (n, %) | 100.0 | 16 (80.0) | 0 (0.0) | |
| ≥3.00 g/g cr (n, %) | 100.0 | 4 (20.0) | 59 (100.0) | |
| UPCR at biopsy (g/g cr) | 100.0 | 1.36 ± 1.00 | 10.16 ± 6.21 | <0.001 |
| <0.30 g/g cr (n, %) | 100.0 | 3 (15.0) | 0 (0.0) | <0.001 |
| 0.30–2.99 g/g cr (n, %) | 100.0 | 17 (85.0) | 0 (0.0) | |
| ≥3.00 g/g cr (n, %) | 100.0 | 0 (0.0) | 59 (100.0) | |

Creatinine before biopsy: The lowest value of serum creatinine 6 months before renal biopsy, DM: Diabetes mellitus, CHD: Coronary heart disease, SBP: Systolic blood pressure, DBP: Diastolic blood pressure, Cr: Creatinine, GFR: Estimated glomerular filtration rate by CKD-EPI equation, AKI: Acute kidney injury based on the lowest creatinine value during the follow-up period, Max UPCR before biopsy: The highest value of UPCR during the 6 months before biopsy, UPCR: Spot urine protein to creatinine ratio (g/g cr).

P-values determined using the Mann-Whitney test, uc: Uncountable.

month. Among the 20 patients in the Non-NS group, seven (35.0%) experienced a UPCR <0.30 g/g creatinine at least once during follow-up. One of seven had recorded UPCRs <0.30, and the other six had recorded 0.3–2.99 g/g creatinine at admission for renal biopsy, respectively. In the Non-NS group, eight out of 20 patients (40.0%) had experienced a relapse of proteinuria, UPCR ≥3.0 g/g creatinine at any point during follow-up, after renal biopsy. These eight patients received steroid treatment without other immunosuppression therapy. One patient was treated with steroids only to address their UPCR of 0.30–2.99 g/g creatinine. The maximum dose of prednisolone administered was 0.92 ± 0.16 (range: 0.5–1.0) mg/kg/day. The remission of proteinuria was achieved in all nine patients. The number of proteinuria relapses was 0.23 ± 0.33 times/year during the follow-up period in the Non-NS group.

Steroids were administered to 93.2% of patients (55/59) in the NS group. The maximum dose of prednisolone administered to the 55 patients in the NS group was 0.88 ± 0.17 (range: 0.4–1.2) mg/kg/day. The proteinuria spontaneously improved to a UPCR <0.3 g/g creatine in four patients. Among these, two patients remained in remission; one increased in UPCR to 0.30–0.29 g/g creatinine, and one experienced a relapse of proteinuria. Thirty-seven out of 55 patients received steroids only, while 14 patients received steroids and calcineurin inhibitors, and four received steroids and cyclophosphamide. Among them, remission of proteinuria was achieved in 51 patients. Eventually, 55 out of 59 patients in the NS group achieved the first remission of proteinuria. Four (6.8%) achieved the first remission without any immunosuppressive treatment. Among 51 patients who used steroids, 34 patients (61.8%) used steroids only, while 13 patients (23.6%) with calcineurin inhibitors, and four patients (7.3%) with cyclophosphamide. Of these, 25 patients ultimately experienced a relapse. The number of proteinuria relapses was 0.25 ± 0.41 times/year during the follow-up period in the NS group. In 10 patients, the proteinuria remained at UPCR <0.3 g/g creatinine after the first remission without relapse throughout the follow-up period. Four patients were treated with steroids and cyclophosphamide. They all head heavy proteinuria over at least 6 g/g at initial manifestation. Podocyte effacement on kidney pathology was diffuse and wide in all four patients. Because they all showed severe nephrotic syndrome, they needed intense immunosuppressive therapy. They achieved the first CR without steroid dependency during withdrawal, and AKI in two developed and improved.

The response rate to steroid treatment in patients with UPCR ≥3.00 g/g creatinine at any point during follow-up was 100% (8/8 patients) in the Non-NS group and 92.3% (51/55 patients) in the NS group (p = 1.000). The frequency of first relapses in patients with UPCR <0.30 g/g creatinine (p = 0.782) and the number of relapses (p = 0.830) did not differ between the groups. At the final visit, the CR rate was 73.4% (58/79), showing no significant difference between the groups (Fig 2 and Table 3). The days to first remission were 10.7 ± 16.5 days in the non-NS group and 8.7 ± 20.2 days in the NS group (p = 0.701) (Fig 3A). Except for nine patients who did not achieve CR, the days to the first relapse were 35.5 ± 49.0 days in the non-NS group and 29.6 ± 43.6 days in the NS group (p = 0.637) (Fig 3B). The eGFR during the observation period was significantly better in the NS group than the Non-NS group because of the higher incidence of AKI at renal biopsy. The ratio of last serum creatinine by the lowest serum creatinine 6 months before the biopsy was 1.47 (± 0.50) in the non-NS group and 1.59 (± 0.82) in the NS group (p = 0.213). The delta of the last serum creatinine to the lowest serum creatinine 6 months before biopsy was 0.32 (± 0.50) in the non-NS group and 0.36 (± 0.60) in the NS group, respectively (p = 0.679). Overall, the incidence rate of renal events, ESRD events, or mortality was not different between the groups (Table 3). Clinical outcomes were not different when analyzing patients divided by the maximal amount of proteinuria 6 months before biopsy (S4 Table).

## Factors associated with the outcomes of MCD

We analyzed the relationship between nephrotic range proteinuria and each event, including AKI, the first CR in patients with UPCR ≥3.00 g/g creatinine at any time, relapse after the first CR, CR at the last visit, and renal events. In the multiple logistic regression analysis, younger age, nephrotic range proteinuria, and lower eGFR at admission were related to AKI's presence at MCD diagnosis. The Cox proportional hazards analysis found that the presence of nephrotic range proteinuria at diagnosis of MCD was not associated with the first remission of proteinuria in patients with UPCR ≥3.00 g/g creatinine at any time, relapse after the first remission of proteinuria, the number of relapses, CR at the last visit, or development of renal events during the observation period (S5 Table).

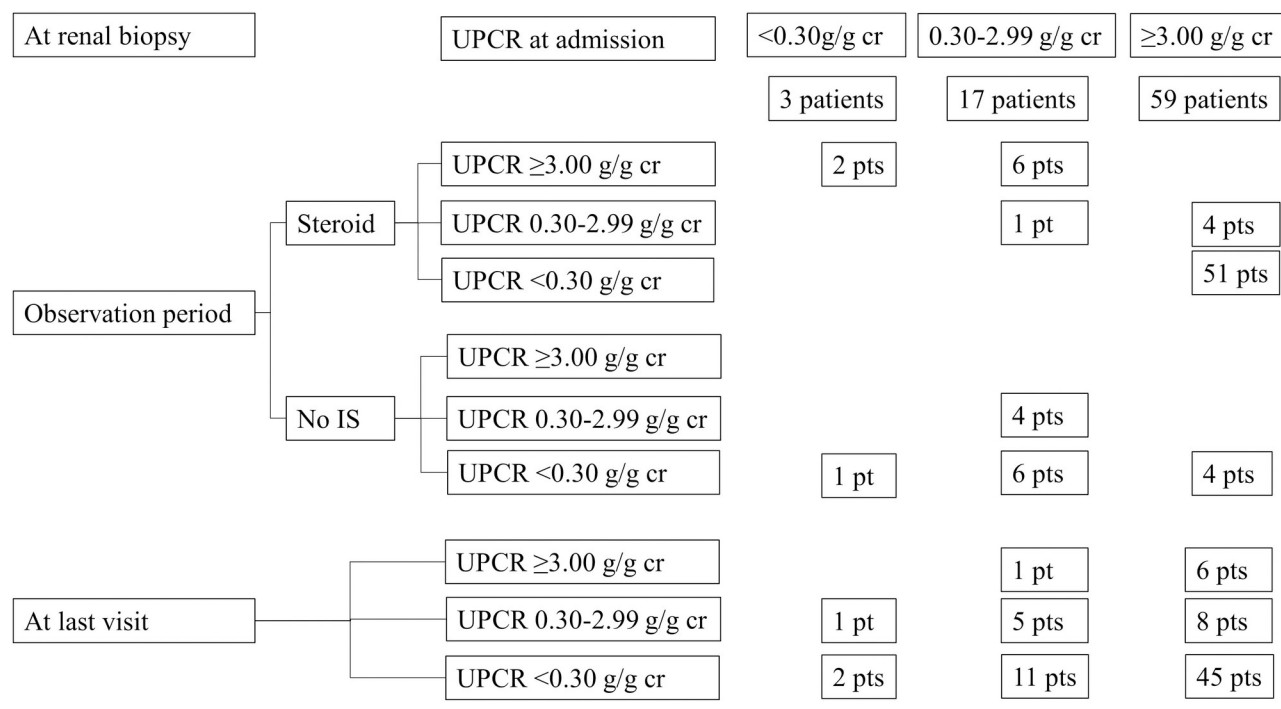

**Fig 2. Progression of proteinuria according to initial amount and treatment.** IS: Immunosuppressive treatment, pt: Patient.

## Discussion

We analyzed the differences in clinical course between nephrotic and non-nephrotic MCD. The long-term outcome of MCD without nephrotic syndrome was rarely reported before. This report focused on the clinical courses during about 5 years of MCD without heavy proteinuria at renal biopsy. The results showed similar remission and relapse rates and response to steroids rates in both groups. On the other hand, AKI was more common in patients with nephrotic syndrome. Interestingly, podocyte effacement on renal pathological examination did not correspond to the amount of proteinuria.

The initial AKI rate differed between the two groups: 59.3% in the NS group compared with just 5.0% in the Non-NS group. Notably, previous reports had found that the incidence of AKI in MCD was 17–38% [10,11], which is lower than that observed in our study. Compared with other studies, we reported AKI in older adults and followed them for longer periods. Furthermore, unlike previous reports, our study's criteria for diagnosing AKI followed the recent guideline. Our study showed similar incidences of relapse and changes in serum creatinine, irrespective of AKI. A previous study in children reported that AKI was unrelated to long-term renal function outcomes [12], despite the children exhibiting severe proteinuria. However, a previous study reported higher relapse rates in patients with acute renal failure than those without. They had defined acute renal failure as a 50% decrease in serum creatinine, which is a stricter criterion than that in our study. Compared with this previous study, we observed more prominent tubular atrophy or interstitial inflammation in patients with MCD with AKI. However, the final serum creatinine assessment during follow-up was not different between the NS and Non-NS groups. This reversible AKI in our study could be explained by glomerular hypoperfusion related to hypoalbuminemia with severe podocyte effacement and decreased slit membrane function [13]. Increased tubular endothelin-1 expression in kidney biopsy samples might be involved in the mechanism of reversible tubular damage in MCD complicated with AKI [14].

**Table 3. Outcomes of minimal change disease.**

| Outcomes | Completeness of data (%) | Non-NS (n = 20) | NS (n = 59) | p-value |
|---|---|---|---|---|
| **FU duration after biopsy (months)** | 100.0 | 58.5 ± 50.3 | 64.3 ± 51.2 | 0.656 |
| **Number of laboratory tests during the follow-up period (/month)** | 100.0 | 0.42 ± 0.23 | 0.72 ± 0.81 | 0.051 |
| **Min. UPCR after biopsy (g/g cr)** | 100.0 | 0.18 ± 0.23 | 0.13 ± 0.27 | 0.464 |
| <0.30 g/g cr (n, %) | 100.0 | 16 (80.0) | 54 (91.5) | 0.220 |
| 0.30–2.99 g/g cr (n, %) | 100.0 | 4 (20.0) | 5 (8.5) | |
| ≥3.00 g/g cr (n, %) | 100.0 | 0 (0.0) | 0 (0.0) | |
| **Max. UPCR after biopsy (g/g cr)** | 100.0 | 4.06 ± 5.10 | 7.99 ± 9.66 | 0.155 |
| <0.30 g/g cr (n, %) | 100.0 | 4 (20.0) | 10 (16.9) | 0.444 |
| 0.30–2.99 g/g cr (n, %) | 100.0 | 8 (40.0) | 16 (27.1) | |
| ≥3.00 g/g cr (n, %) | 100.0 | 8 (40.0) | 33 (55.9) | |
| **UPCR at last visit (g/g cr)** | 100.0 | 0.75 ± 1.58 | 1.34 ± 4.60 | 1.000 |
| <0.30 g/g cr (n, %) | 100.0 | 13 (65.0) | 45 (76.3) | 0.225 |
| 0.30–2.99 g/g cr (n, %) | 100.0 | 6 (30.0) | 8 (13.6) | |
| ≥3.00 g/g cr (n, %) | 100.0 | 1 (5.0) | 6 (10.2) | |
| **Creatinine at last visit (mg/dl)** | 100.0 | 0.97 ± 0.62 | 1.02 ± 0.66 | 0.215 |
| **GFR at last visit (ml/min/1.73 m2)** | 100.0 | 87 ± 26 | 85 ± 31 | 0.698 |
| **Treatment after renal biopsy** | | | | |
| RAS blocker (n, %) | 100.0 | 12 (60.0) | 31 (52.5) | 0.612 |
| Anti-hypertensive medication (n, %) | 100.0 | 13 (65.0) | 41 (69.5) | 0.783 |
| Anti-diabetic medication (n, %) | 100.0 | 6 (30.0) | 23 (39.0) | 0.595 |
| **Treatment for induction of the first CR** | | | | |
| Immunosuppression (n, %) | 100.0 | 9 (45.0) | 55 (93.2) | <0.001 |
| Steroid only (n, %) | 100.0 | 9 (45.0) | 37 (62.7) | |
| Steroid and calcineurin inhibitor (n, %) | 100.0 | 0 (0.0) | 14 (23.7) | |
| Steroid and cyclophosphamide (n, %) | 100.0 | 0 (0.0) | 4 (6.8) | |
| The highest dose of prednisolone (mg/kg/day) | 100.0 | 0.92 ± 0.16 | 0.88 ± 0.17 | 0.225 |
| The total dose of prednisolone until CR or 1st relapse (mg) | 100.0 | 5635 ± 2256 | 7014 ± 3592 | 0.284 |
| **Outcomes** | | | | |
| The first CR (n, %) among patients with UPCR ≥3.00 g/g cr (n, %) | 100.0 | 8/8 (100.0) | 55/59 (93.2) | 1.000 |
| The first relapse among patients with UPCR <0.30 g/g cr (n, %) | 100.0 | 8/16 (50.0) | 25/55 (45.5) | 0.782 |
| SD at the first relapse (n, %) | 100.0 | 1/16 (6.3) | 10/55 (18.2) | 0.436 |
| Number of relapses after biopsy (/year) | 100.0 | 0.23 ± 0.33 | 0.25 ± 0.41 | 0.830 |
| CR at last visit (n, %) | 100.0 | 13 (65.0) | 45 (76.3) | 0.384 |
| The slope of GFR (ml/min/1.73 m2/year) | 100.0 | (-)4.62 ± 8.06 | 3.64 ± 13.2 | 0.001 |
| Renal events during follow-up (n, %) | 100.0 | 3 (15.0) | 16 (27.1) | 0.371 |
| ESRD during follow-up (n, %) | 100.0 | 1 (5.0) | 9 (15.3) | 0.438 |
| Death during follow-up (n, %) | 100.0 | 0 (0.0) | 4 (6.8) | 0.567 |

Min UPCR after biopsy: The lowest value of UPCR during the follow-up period starting 1 month after biopsy, Max UPCR after biopsy: The highest value of UPCR during the follow-up period starting 1 month after biopsy, FU duration after biopsy: Follow-up duration between renal biopsy and the last test of UPCR, RAS: Renin-angiotensin-system, CR: Complete remission of proteinuria <0.3 g/g creatinine, relapse: UPCR >3.0 g/g creatinine after achieving CR of UPCR, SD: The presence of steroid dependency in a patient where relapse of UPCR occurred during steroid tapering or within 2 weeks after cessation of steroids, renal event: Any decrease of GFR of more than 50% during a follow-up visit compared with that at renal biopsy, GFR <15 ml/min/1.73 m$^2$, or development of ESRD, The first CR (n, %) among patients with UPCR >3.00 g/g cr: In the Non-NS group, patients who had increased proteinuria of >3.00 g/g cr during the observation period. The first relapse among patients with UPCR <0.30 g/g cr: In the Non-NS group, patients who had UPCR of <0.30 g/g cr at any time of period.

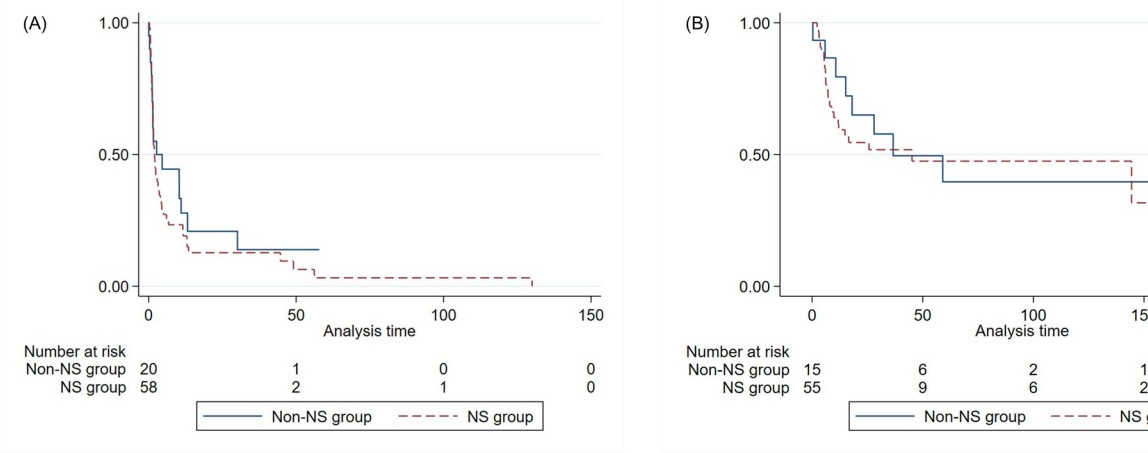

**Fig 3.** Survival time from the kidney biopsy to the first remission (A) and time from the first remission to the first relapse (B). The linear line indicates the Kaplan-Meier graph line in the Non-NS group. The dashed line indicates that in the NS group.

The clinical course, including remission and relapse, did not differ between patients in both groups. The final CR rate was also equivalent in the NS and Non-NS groups. In a previous follow-up period similar to our study [15], the CR rate at the final follow-up visit was 87% in the NS group, compared to the 73.4% observed in our study. During the follow-up period, there was no difference in the rates of first CR and development of first relapse, regardless of the presence of nephrotic syndrome at the time of kidney biopsy. The eGFR decreased to a greater extent in the Non-NS group, although the mean levels of eGFR and serum creatinine at the last visit were similar in the two groups. Patients in the Non-NS group had better renal function and higher serum albumin levels on the biopsy day than those in the NS group. Half of the patients that did not have nephrotic syndrome received steroids after developing proteinuria during follow-up. All patients in the Non-NS group who received steroids achieved CR. Spontaneous remission from nephrotic proteinuria before admission was observed in one patient. Four of the 59 patients with nephrotic syndrome achieved CR spontaneously without immunosuppressive agents. Notably, previous studies reported spontaneous proteinuria remission in 8–33% of untreated patients with MCD [16,17]. In one study, half of the patients who received no immediate immunosuppressive treatment achieved spontaneous remission, showing delayed amelioration of nephrotic syndrome.

Two patients in the NS group and one in the Non-NS group showed no podocyte effacement. Similar to our study, the severity of podocyte effacement was not correlated with the prognosis of MCD in previous reports [18]. However, others have found a positive correlation between the amount of proteinuria and the severity of podocyte effacement [19,20]. These studies included several types of glomerulonephritis besides MCD, while ours focused on a relatively large number of MCD patients. Sampling error could be another cause for the discrepancy in findings, where samples may have been obtained from regions distant from the location of podocyte effacement.

On the other hand, this unclear association between podocyte effacement and the severity of proteinuria might suggest the presence of unknown mechanisms in MCD. Proteinuria in MCD is usually induced by T-cell dysfunction and, eventually, a B-cell pathway [21]. In addition, spontaneous relief of heavy proteinuria in MCD cannot be explained by conventional mechanisms, such as enlarged slit pores or a negative charge in the glomerular basement membrane. Furthermore, several recent studies have presented putative molecular mechanisms involving angiopoietin like-4, interleukin-8, and hemopexin [22].

There are some limitations to this study. This was a retrospective study, and the confounders were not strictly controlled. Therefore, we only included patients without previous immunosuppressive treatment. In addition, using strict inclusion criteria meant that many patients were excluded from the study. As a result, the number of patients included was small. Second, only adults were included in this study, despite MCD being prevalent in children. However, the clinical course of MCD differs in various aspects between adults and children. Therefore, it was meant to describe the prognosis of MCD in adults specifically. In the same sense, further studies considering the initial amount of proteinuria are required for patients of all ages. Third, the pathologic findings in some patients were away from classic MCD in our study. Findings were EM findings, such as < 50% of podocyte effacement, and IM findings, such as mild deposits. Tubulointerstitial damage to differentiate between MCD and unsampled FSGS was not thoroughly investigated. This might be because MCD is sometimes manifested atypically. However, we tried sampling adequate glomeruli to maintain sampling adequacy. Our study suggested no definite correlation between pathologic severity and clinical features such as the amount of proteinuria; however, we still need further findings to prevail this.

In conclusion, there was no overall clinicopathological difference in the prognosis of MCD between patients with and without heavy proteinuria. Therefore, patients who show non-nephrotic proteinuria as an initial manifestation should be monitored as closely as those with nephrotic MCD without spontaneous remission. Future studies involving recently developed immunosuppressive treatments might help elucidate the mechanisms underlying MCD relapse.

## Supporting information

**S1 Table. Characteristics of patients with minimal change disease according to the highest amount of proteinuria during 6 months before renal biopsy.** Creatinine before biopsy: The lowest value of serum creatinine 6months before renal biopsy, DM: Diabetes mellitus, CHD: Coronary heart disease, SBP: Systolic blood pressure, DBP: Diastolic blood pressure, Cr: Creatinine, GFR: Estimated glomerular filtration rate by CKD-EPI equation, AKI: Acute kidney injury based on the lowest creatinine value during the follow-up period, Max UPCR before biopsy: The highest value of UPCR during the 6 months before biopsy, UPCR: Spot urine protein to creatinine ratio (g/g cr).
(DOCX)

**S2 Table. Renal pathologic findings of patients according to the amount of proteinuria.** LM findings: Light microscopy findings, Mes: Mesangial, IF findings: Immunofluorescent microscopy findings, EM findings: Electron microscopy findings P-values determined using the Mann-Whitney test, uc: Uncountable.
(DOCX)

**S3 Table. Effect of podocyte effacement severity on proteinuria.** For the amount of proteinuria, a multiple linear regression model that adjusted for age, sex, blood pressures, and pathologic findings of podocyte effacement (none, focal, and diffuse) was used. For the presence of NS, a multiple logistic regression model that adjusted for age, sex, serum glucose, pathologic findings of interstitial fibrosis, interstitial inflammation, tubular atrophy, and podocyte effacement (none, focal, and diffuse) was used. NS: Nephrotic-range proteinuria, CI: Confidence interval, RR, relative risk.
(DOCX)

**S4 Table. Outcomes of minimal change disease according to the highest amount of proteinuria during 6 months before renal biopsy.** Min UPCR after biopsy: The lowest value of

UPCR during the follow-up period starting 1 month after biopsy, Max UPCR after biopsy: The highest value of UPCR during the follow-up period starting 1 month after biopsy, FU duration after biopsy: Follow-up duration between renal biopsy and the last test of UPCR, RAS: Renin-angiotensin-system, CR: Complete remission of proteinuria <0.3 g/g creatinine, relapse: UPCR >3.0 g/g creatinine after achieving CR of UPCR, SD: Presence of steroid dependency in a patient where relapse of UPCR occurred during steroid tapering or within 2 weeks after cessation of steroids, renal event: Any decrease of GFR of more than 50% during a follow-up visit compared to that at renal biopsy, GFR <15 ml/min/1.73 m$^2$, or development of ESRD, The first CR (n, %) among patients with UPCR >3.00 g/g cr: In the Non-NS group, patients who had increased proteinuria of >3.00 g/g cr during observation period. The first relapse among patients with UPCR <0.30 g/g cr: In the Non-NS group, patients who had UPCR of <0.30 g/g cr at any time of period.
(DOCX)

**S5 Table. Effect of presence of nephrotic range proteinuria at diagnosis of MCD on each event.** For presence of AKI at diagnosis, a multiple logistic regression model adjusted for age, sex, serum albumin, eGFR, blood pressures, and severity of podocyte effacement was used. For the first remission in patients with UPCR >3.0 g/g cr: A Cox proportional hazards model adjusted for age; sex; and pathologic findings of deposition of IgA, IgG, lambda chains, or interstitial inflammation, was used. For the first relapse in patients with UPCR <0.3 g/g cr: A Cox proportional hazards model adjusted for age, sex, and steroid treatment was used. For the number of relapses in patients with UPCR <0.3 g/g cr: A multiple linear regression model adjusted for age, sex, serum cholesterol, and steroid treatment was used. For renal events, a Cox proportional hazards model adjusted for age, sex, presence of AKI at renal biopsy, presence of hypertension, eGFR, pathologic deposition of IgA, RAS blockade medication, antihypertensive drugs, anti-diabetic drugs, immunosuppressive drugs, and number of relapses of nephrotic range proteinuria was used. For the remission of proteinuria at the last visit, a Cox proportional hazards model adjusted for age; sex; serum albumin; presence of diabetes mellitus; first remission of proteinuria; relapse of proteinuria; and pathologic findings of changes in the mesangial matrix, interstitial inflammation, and presence of atherosclerosis, was used. AKI: Acute kidney injury, eGFR: Estimated glomerular filtration rate, cr: Creatinine, RAS: Renin-angiotensin system, UPCR: Urine protein/creatinine ratio, MCD: Minimal change disease.
(DOCX)

**S1 File. A minimal anonymized data set.**
(ZIP)

## Author Contributions

**Conceptualization:** Ho Jun Chin.

**Data curation:** Giae Yun, Eun-Jeong Kwon, Jin Ho Paik.

**Formal analysis:** Ho Jun Chin.

**Investigation:** Seokwoo Park, Jin Ho Paik.

**Methodology:** Jong Cheol Jeong.

**Software:** Jong Cheol Jeong.

**Supervision:** Sejoong Kim, Ki Young Na.

**Visualization:** Hyung Eun Son.

**Writing – original draft:** Hyung Eun Son.

**Writing – review & editing:** Ho Jun Chin.

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
