## [Decision Letter · Decision Letter 0]

28 Oct 2022

PONE-D-22-24683The outcome of minimal change disease without nephrotic syndromePLOS ONE

Dear Dr. Chin,

Thank you for submitting your manuscript to PLOS ONE. After careful consideration, we feel that it has merit but does not fully meet PLOS ONE’s publication criteria as it currently stands. Therefore, we invite you to submit a revised version of the manuscript that addresses the points raised during the review process.

We look forward to receiving your revised manuscript.

Kind regards,

Maria Lourdes Gonzalez Suarez, MD, PhD

Academic Editor

PLOS ONE

Journal Requirements:

"NO"

"NO"

Additional Editor Comments:

Please address reviewers comments

Reviewers' comments:

Reviewer's Responses to Questions

**Comments to the Author**

1. Is the manuscript technically sound, and do the data support the conclusions?

Reviewer #1: Yes

Reviewer #2: Partly

2. Has the statistical analysis been performed appropriately and rigorously? 

Reviewer #1: Yes

Reviewer #2: Yes

3. Have the authors made all data underlying the findings in their manuscript fully available?

Reviewer #1: Yes

Reviewer #2: Yes

4. Is the manuscript presented in an intelligible fashion and written in standard English?

Reviewer #1: Yes

Reviewer #2: Yes

5. Review Comments to the Author

Reviewer #1: - This is a well done study overall, with the limitations that a retrospective study has, as it is pointed out in the Discussion.

- The methods, data and results are well presented and explained.

- Some patients in the NS group received cyclophosphamide- it would be good to discuss how this could have impacted the results.

- One of the definition of renal events was decrease in eGF by 50% at a follow-up visit compared to that at renal biopsy. Perhaps it would have been better to compare to the baseline Cr (or average Cr in weeks/few months prior to kidney biopsy). Because patients could have been pre-renal or have any other cause of AKI that was not really related to MN, and this could have biased the results.

- It would be great to mention if, among the patients who were on dialysis, how many of them were on dialysis at time of biopsy, since the Creatinine would be falsely low, and thus, biasing the end point "decrease in GFR by 50%).

- Is there a reason why Highest dose of prednisolone was used to compare groups as opposed to average dose of prednisolone?

- Did any patients have any findings of diabetic nephropathy in addition to the MN findings which could have impacted the results?

Reviewer #2: The novelty of this study is that it describes the long-term prognosis of MCD without nephrotic syndrome in adults and compares this to a group of patients with MCD and nephrotic syndrome

However, I believe that there is a selection bias since the study is including some patients who present features that are not typical of classic MCD, such as deposition of immunoglobulins or complement on immunofluorescence or focal foot process effacement (classic MCD presents with diffuse foot process effacement). Also, notably a large number of patients had concomitant interstitial inflammation which I found unusual and to be a large confounder since this finding could also explain AKI or progression to ESRD independently of the glomerular disease.

Therefore, I believe that since a large proportion of this cohort does not have the classic MCD histology, it is unclear if the diagnosis was correct and therefore if the conclusion suggested by the authors is applicable

Review Comments

I would suggest to use a Kaplan Meier curve to present the comparison between groups in regards to time to complete remission after treatment and time to first relapse

Please find below specific comments in regards to the manuscript content.

Lines 85-88

We defined MCD pathologically as near normal findings on light microscopy, except for mild expansion of the mesangium, and no electron deposit and focal or diffuse podocyte effacement on electron microscopy

Comment

I believe the definition of MCD used could be more descriptive, for example:

“Near normal findings on light microscopy, except for mild expansion of the mesangium or global glomerulosclerosis, which is a nonspecific finding that can also be seen. There is absence of segmental sclerosis. There are no complement or immunoglobulin deposits on immunofluorescence microscopy. On electron microscopy there is diffuse effacement of podocyte foot processes. There is absence of electron dense deposits or thickening of the glomerular basement membrane.

Note. The characteristic histologic lesion in MCD is diffuse effacement (>50-80%) of the epithelial foot processes on electron microscopy. The proposed diagnosis of MCD on this manuscript includes patients with focal or diffuse foot process effacement. Limited foot process effacement (<50%) suggests the process is probably not MCD, unless the patient has been treated with partial response. However, on line 90 it is mentioned that patients who had immunosuppressive treatment before renal biopsy were excluded.

I suggest to incorporate in the definition of MCD the exclusion of segmental sclerosis. Focal foot process effacement is more commonly associated with FSGS.

Lines 129-131

As we had previously described [10], glomerular lesions such as global sclerosis, segmental sclerosis, glomerular ischemic change, and crescentic changes were reported as a proportion of the total number of glomeruli in an evaluated specimen

Comment

I found this statement confusing. On lines 85-88 it is mentioned that MCD was defined as near normal findings on light microscopy. But here it states that segmental sclerosis, glomerular ischemic change, crescentic changes are reported. If these findings are present, it would not be consistent with a diagnosis of MCD.

Lines 138-139

The findings assessed using electron microscopy included presence of electron dense deposits in the area of the mesangium, subendothelium, and subepithelium

Comment

I found this statement confusing. On line 87 you mention that one of the criteria to define MCD was absence of electron dense deposits. But here you mention that you are evaluating electron dense deposits based on location. If electron dense deposits are present this would NOT be consistent with a diagnosis of MCD

Lines 140-141

severity of foot process effacement of the podocytes, which were reported as none, focal (mild, moderate, moderate to severe, severe), or diffuse.

Comment

I believe that if there is only mild to moderate foot process effacement it is possible that the diagnosis of MCD is incorrect

Lines 187-188

The patients were divided into Non-NS (n=20, 25.3%) and NS-groups (n=59, 74.7%) according to the presence of proteinuria ≥ 3g/g on the day of biopsy

Comment

Table 2 shows that 4 out of 20 patients in the non-NS group had a max UPCR ≥3.00 g/g cr before biopsy. However, on the day of the biopsy all patients on the non-NS group already had a UPCR of ≤ 3.00 g/g cr. This means that in some patients with MCD and nephrotic proteinuria the amount of proteinuria was already spontaneously decreasing without therapy.

I suggest to classify these patients in the nephrotic syndrome group as opposed to the non-NS group.

Maybe you can refer to both the max UPCR before or at the time of kidney biopsy to stratify patients in each one the groups

Lines 198-199

Serum levels of protein and albumin were lower in the NS group, as were the levels of total cholesterol (p<0.001, table 2)

Comment

Table 2 shoes that cholesterol levels were higher in the NS group which is the opposite of what is mentioned on the text. As expected, patients with nephrotic syndrome had higher cholesterol levels

Lines 207-208

Interestingly, two patients in the NS group and one in the Non-NS group showed no podocyte effacement.

Comment

Please clarify this. The definition for MCD proposed mentions that patients should have focal or diffuse foot process effacement. If these patients did not have podocyte effacement, how were they diagnosed with MCD?

Lines 209-210 Table 2

Comment

Table 2 shows on immunofluorescence that there is staining for IgG, IgA, C3 or C1q in multiple patients.

Classic MCD does not have immunoglobulin or complement deposits. There may be variants of MCD (eg, IgM nephropathy or C1q nephropathy). But it is unclear if this represent separate conditions. Moreover, the presence of Ig M deposits in a patient with MCD has been associated with a poorer prognosis and a lower response to glucocorticoids compared to patients with MCD without deposits

Moreover, the proportion of patients with diffuse process effacement in the non-NS group is low (40%) which in my opinion raises questions about whether they had the correct diagnosis.

6. PLOS authors have the option to publish the peer review history of their article (what does this mean?). If published, this will include your full peer review and any attached files.

Reviewer #1: **Yes: **Aldo Torres Ortiz

Reviewer #2: **Yes: **Efren Alejandro Chavez Morales

---

## [Author Response · Author response to Decision Letter 0]

29 Jan 2023

Reviewer #1: 

We really appreciate your recommendations. 

Answers to your each recommendation are as below. 

Comment>

- Some patients in the NS group received cyclophosphamide

- it would be good to discuss how this could have impacted the results.

Answer>

Thank you for recommendation.

We added sentences as below (12p). 

“4 patients were treated with steroid and cyclophosphamide. They all head heavy proteinuria over at least 6 g/g at initial manifestation. Podocyte effacement on kidney pathology was diffused and wide in all 4 patients. Because they all showed severe nephrotic syndrome, they needed intense immunosuppressive therapy. They achieved the first CR without steroid dependency during withdrawal, and AKI, developed in 2 of them, was relieved.” (12p)

Comment>

- One of the definitions of renal events was decrease in eGFR by 50% at a follow-up visit compared to that at renal biopsy. Perhaps it would have been better to compare to the baseline Cr (or average Cr in weeks/few months prior to kidney biopsy). Because patients could have been pre-renal or have any other cause of AKI that was not really related to MN, and this could have biased the results.

Answer>

Thank you for kind recommendation. 

We had also collected lowest creatinine during 6 months before renal biopsy. Serum creatinine was a little higher in non-NS group than in NS group without statistical significance. However, it did not show difference between two groups in an aspect of clinical outcome with serum creatinine. We added this to result section as below. (16p) 

“The response rate to steroid treatment in patients with UPCR ≥3.00 g/g creatinine at any point during follow-up was 100% (8/8 patients) in the Non-NS group and 92.3% (51/55 patients) in the NS group (p=1.000). The frequency of first relapses in patients with UPCR <0.30 g/g creatinine (p=0.782) and the number of relapses (p=0.830) did not differ between the groups. At the final visit, the CR rate was 73.4% (58/79), showing no significant difference between the groups (Fig 2 and Table 3). The days to the first remission were 10.7 ± 16.5 days in non-NS group and 8.7 ± 20.2 days in NS group (p=0.701) (Fig 3-A). Except 9 patients who did not achieve CR, the days to the first relapse was 35.5 ± 49.0 days in non-NS group and 29.6 ± 43.6 days in NS group (p=0.637) (Fig 3-B). The eGFR during the observation period was significantly better in the NS group compared to the Non-NS group because of higher incidence of AKI at renal biopsy in the NS group. The ratio of last serum creatinine by the lowest serum creatinine 6 months before biopsy was 1.47 (± 0.50) in non-NS group, when was 1.59 (± 0.82) in NS group (p=0.213). The delta of last serum creatinine to the lowest serum creatinine 6 months before biopsy was 0.32 (± 0.50) in non-NS group and 0.36 (± 0.60) in NS group, respectively (p = 0.679). Overall, The incidence rate of renal events, ESRD events, or mortality was not different between the groups (Table 3).” (16p)

Comment>

- It would be great to mention if, among the patients who were on dialysis, how many of them were on dialysis at time of biopsy, since the Creatinine would be falsely low, and thus, biasing the end point "decrease in GFR by 50%).

Answer>

I appreciate for your comment.

Five patients started hemodialysis due to acute kidney injury when biopsy was done. However, because there were at least more than 3 days between the day of biopsy and the day of starting hemodialysis, all baseline creatinine was measured before starting dialysis. 

Comment>

- Is there a reason why Highest dose of prednisolone was used to compare groups as opposed to average dose of prednisolone?

Answer>

Thank you for recommendation. 

We wanted to emphasize on the similar dose of steroid between two groups. So, we suggested both the highest dose of steroid, mostly the start dose, and the total dose of steroid together. As we know, although there is standard regimen for MCD treatment, low-dose steroid regimen or short-term use of steroid would be possible in some situations. In this study, patients in two groups used similar dose of steroid regardless of initial among of proteinuria, once steroids were used.

Comment>

- Did any patients have any findings of diabetic nephropathy in addition to the MN findings which could have impacted the results?

Answer>

I appreciate for your recommendation. 

We reviewed our data again in an aspect of diabetic nephropathy. Although there were 15 % of patients (12 patients) who were diagnosed with diabetes mellitus, diabetic nephropathy was exclusive diagnosis. Because we excluded the collection of data who were suspected as diabetic nephropathy clinically or pathologically in methodology.

 

Reviewer #2.

Thank you for your thoughtful recommendations. 

Sentences below are my answers to your comments.

Comment>

- I would suggest to use a Kaplan Meier curve to present the comparison between groups in regards to time to complete remission after treatment and time to first relapse

Answer>

We appreciate for your recommendations.

Kaplan-Meier graphs for the 1st remission and 1st relapse was added to the manuscript as below (figure 3-A and 3-B).Because as the mean times were not different between two groups, the description was added as below (16p)

“The response rate to steroid treatment in patients with UPCR ≥3.00 g/g creatinine at any point during follow-up was 100% (8/8 patients) in the Non-NS group and 92.3% (51/55 patients) in the NS group (p=1.000). The frequency of first relapses in patients with UPCR <0.30 g/g creatinine (p=0.782) and the number of relapses (p=0.830) did not differ between the groups. At the final visit, the CR rate was 73.4% (58/79), showing no significant difference between the groups (Fig 2 and Table 3). The days to the first remission were 10.7 ± 16.5 days in non-NS group and 8.7 ± 20.2 days in NS group (p=0.701) (Fig 3-A). Except 9 patients who did not achieve CR, the days to the first relapse was 35.5 ± 49.0 days in non-NS group and 29.6 ± 43.6 days in NS group (p=0.637) (Fig 3-B). The eGFR during the observation period was significantly better in the NS group compared to the Non-NS group because of higher incidence of AKI at renal biopsy in the NS group. The ratio of last serum creatinine by the lowest serum creatinine 6 months before biopsy was 1.47 (± 0.50) in non-NS group, when was 1.59 (± 0.82) in NS group (p=0.213). The delta of last serum creatinine to the lowest serum creatinine 6 months before biopsy was 0.32 (± 0.50) in non-NS group and 0.36 (± 0.60) in NS group, respectively (p = 0.679). Overall, The incidence rate of renal events, ESRD events, or mortality was not different between the groups (Table 3).” (16p)

Comment>

- Please find below specific comments in regards to the manuscript content.

[Lines 85-88]

We defined MCD pathologically as near normal findings on light microscopy, except for mild expansion of the mesangium, and no electron deposit and focal or diffuse podocyte effacement on electron microscopy

Comment>

I believe the definition of MCD used could be more descriptive, for example:

“Near normal findings on light microscopy, except for mild expansion of the mesangium or global glomerulosclerosis, which is a nonspecific finding that can also be seen. There is absence of segmental sclerosis. There are no complement or immunoglobulin deposits on immunofluorescence microscopy. On electron microscopy there is diffuse effacement of podocyte foot processes. There is absence of electron dense deposits or thickening of the glomerular basement membrane.

Note. The characteristic histologic lesion in MCD is diffuse effacement (>50-80%) of the epithelial foot processes on electron microscopy. The proposed diagnosis of MCD on this manuscript includes patients with focal or diffuse foot process effacement. Limited foot process effacement (<50%) suggests the process is probably not MCD, unless the patient has been treated with partial response. However, on line 90 it is mentioned that patients who had immunosuppressive treatment before renal biopsy were excluded.

I suggest to incorporate in the definition of MCD the exclusion of segmental sclerosis. Focal foot process effacement is more commonly associated with FSGS.

Answer>

Thank you for your kind recommendation.

The sentences were corrected as you suggested as below (6p in the manuscript) 

“We defined MCD pathologically as near normal findings on light microscopy, except for mild expansion of the mesangium or global glomerulosclerosis which could also be seen as nonspecific findings. Segmental sclerosis was not seen. There were no complement or immunoglobulin deposits on immunofluorescence microscopy. Diffuse effacement of podocyte foot processes was seen on electron microscopy.” (6p) 

[Lines 129-131]

As we had previously described [10], glomerular lesions such as global sclerosis, segmental sclerosis, glomerular ischemic change, and crescentic changes were reported as a proportion of the total number of glomeruli in an evaluated specimen

Comment>

I found this statement confusing. On lines 85-88 it is mentioned that MCD was defined as near normal findings on light microscopy. But here it states that segmental sclerosis, glomerular ischemic change, crescentic changes are reported. If these findings are present, it would not be consistent with a diagnosis of MCD.

Answer>

I appreciate for your comment. 

Because we intended to descript the pathological diagnostic steps to diagnose MCD, “renal pathology” section might have mentioned all pathological descriptions in general. To clarify our intend, we corrected sentences as below . And we also abbreviated table 2 and added details of pathologic findings in supplementary table 1. 

“As we had previously described [10], pathologic diagnosis contains following contents. Glomerular lesions such as global sclerosis, segmental sclerosis, glomerular ischemic change, and crescentic changes were reported as a proportion of the total number of glomeruli in an evaluated specimen. A semi-quantitative assessment of changes in mesangial cellularity, mesangial matrix, tubular atrophy, and interstitial inflammation and fibrosis was performed, with the results classified as normal, mild, moderate, moderate to severe, and severe. Vascular abnormality was defined as the presence of arteriolar hyalinosis and arteriosclerosis. The immunofluorescence study was performed using the classic direct technique with antibodies against 8 proteins (IgG, IgM, IgA, C3, C1q, fibrinogen, kappa and lambda chains). The results were reported semi-quantitatively as negative (0), trace (0.5) and 1–3 positive (1-3). The findings by electron microscopy were descripted as followings; presence of electron dense deposits in the area of the mesangium, subendothelium, and subepithelium; and severity of foot process effacement of the podocytes, which were reported as none, focal (mild, moderate, moderate to severe, severe), or diffuse.” (8p in the manuscript)

[Lines 138-139]

The findings assessed using electron microscopy included presence of electron dense deposits in the area of the mesangium, subendothelium, and subepithelium

Comment>

I found this statement confusing. On line 87 you mention that one of the criteria to define MCD was absence of electron dense deposits. But here you mention that you are evaluating electron dense deposits based on location. If electron dense deposits are present this would NOT be consistent with a diagnosis of MCD

Answer>

I am really appreciate for your comment. 

As you said, it should be not descriptive of MCD. We just wanted to descript reviewed pathologic findings in detail. We found that there were no electron deposits as we expected. To clarify our intends, we fixed sentences as below. (8p) And we also abbreviated table 2 and added details of pathologic findings in supplementary table 1.

“As we had previously described [10], pathologic diagnosis contains following contents. Glomerular lesions such as global sclerosis, segmental sclerosis, glomerular ischemic change, and crescentic changes were reported as a proportion of the total number of glomeruli in an evaluated specimen. A semi-quantitative assessment of changes in mesangial cellularity, mesangial matrix, tubular atrophy, and interstitial inflammation and fibrosis was performed, with the results classified as normal, mild, moderate, moderate to severe, and severe. Vascular abnormality was defined as the presence of arteriolar hyalinosis and arteriosclerosis. The immunofluorescence study was performed using the classic direct technique with antibodies against 8 proteins (IgG, IgM, IgA, C3, C1q, fibrinogen, kappa and lambda chains). The results were reported semi-quantitatively as negative (0), trace (0.5) and 1–3 positive (1-3). The findings by electron microscopy were descripted as followings; presence of electron dense deposits in the area of the mesangium, subendothelium, and subepithelium; and severity of foot process effacement of the podocytes, which were reported as none, focal (mild, moderate, moderate to severe, severe), or diffuse.” (8p)

[Lines 140-141]

severity of foot process effacement of the podocytes, which were reported as none, focal (mild, moderate, moderate to severe, severe), or diffuse.

Comment>

I believe that if there is only mild to moderate foot process effacement it is possible that the diagnosis of MCD is incorrect

Answer>

Thank you for your comment. 

As you said, it should be not descriptive of MCD. We just wanted to descript reviewed pathologic findings in detail. To clarify our intends, we fixed sentences as below. (8p in the manuscript). And we also abbreviated table 2 and added details of pathologic findings in supplementary table 1.

“As we had previously described [10], pathologic diagnosis contains following contents. Glomerular lesions such as global sclerosis, segmental sclerosis, glomerular ischemic change, and crescentic changes were reported as a proportion of the total number of glomeruli in an evaluated specimen. A semi-quantitative assessment of changes in mesangial cellularity, mesangial matrix, tubular atrophy, and interstitial inflammation and fibrosis was performed, with the results classified as normal, mild, moderate, moderate to severe, and severe. Vascular abnormality was defined as the presence of arteriolar hyalinosis and arteriosclerosis. The immunofluorescence study was performed using the classic direct technique with antibodies against 8 proteins (IgG, IgM, IgA, C3, C1q, fibrinogen, kappa and lambda chains). The results were reported semi-quantitatively as negative (0), trace (0.5) and 1–3 positive (1-3). The findings by electron microscopy were descripted as followings; presence of electron dense deposits in the area of the mesangium, subendothelium, and subepithelium; and severity of foot process effacement of the podocytes, which were reported as none, focal (mild, moderate, moderate to severe, severe), or diffuse.” (8p in the manuscript)

[Lines 187-188]

The patients were divided into Non-NS (n=20, 25.3%) and NS-groups (n=59, 74.7%) according to the presence of proteinuria ≥ 3g/g on the day of biopsy

Comment>

Table 2 shows that 4 out of 20 patients in the non-NS group had a max UPCR ≥3.00 g/g cr before biopsy. However, on the day of the biopsy all patients on the non-NS group already had a UPCR of ≤ 3.00 g/g cr. This means that in some patients with MCD and nephrotic proteinuria the amount of proteinuria was already spontaneously decreasing without therapy.

I suggest to classify these patients in the nephrotic syndrome group as opposed to the non-NS group.

Maybe you can refer to both the max UPCR before or at the time of kidney biopsy to stratify patients in each one the groups

Answer>

We are thankful for your recommendation. 

Comparing two groups divided by maximal amount of proteinuria during 6 months before renal biopsy was done. It showed similar results with previous comparison with the amount of proteinuria on the day of renal biopsy. This result was added to the supplementary materials (S1 Table, S2 Table) and descripted in the manuscript as below. 

“4 patients out of 20 patients who belonged to Non-NS group had a max UPCR ≥ 3g/g creatinine before biopsy. When dividing patients according to the maximal amount of proteinuria 6 months before biopsy by 3g/g creatinine, 16 patients had < 3 g/g creatinine of proteinuria and the other 63 patients had over 3g/g creatinine of proteinuria. When comparing groups according to a max UPCR before biopsy, the two groups showed high UPCR, lower GFR, and higher proportion of AKI (S1 Table).” (12p in the manuscript)

” These trend of clinical outcome was not different when analyzing patient divided by the maximal amount of proteinuria during 6 months before biopsy (S4 Table).” (18p in the manuscript)

[Lines 198-199]

Serum levels of protein and albumin were lower in the NS group, as were the levels of total cholesterol (p<0.001, table 2)

Comment>

Table 2 shoes that cholesterol levels were higher in the NS group which is the opposite of what is mentioned on the text. As expected, patients with nephrotic syndrome had higher cholesterol levels

Answer>

I really appreciate your comment. Description in lines you mentioned should be corrected as below. (12p in the manuscript)

“. Serum levels of protein and albumin were lower in the NS group, when the level of total cholesterol was higher (p <0.001, Table 2).” (12p in the manuscript)

[Lines 207-208]

Interestingly, two patients in the NS group and one in the Non-NS group showed no podocyte effacement.

Comment>

Please clarify this. The definition for MCD proposed mentions that patients should have focal or diffuse foot process effacement. If these patients did not have podocyte effacement, how were they diagnosed with MCD?

Answer>

Thank you for your comment. The sentences refer to 2 patients who are suspected to have had MCD change before biopsy. As we mentioned in discussion, there are controversies on the correlation between the amount of proteinuria and the severity of podocyte effacement. Although we showed higher relative risk on the podocyte effacement and the amount of proteinuria, it did not consistent . 

[Lines 209-210] Table 2

Comment>

Table 2 shows on immunofluorescence that there is staining for IgG, IgA, C3 or C1q in multiple patients.

Classic MCD does not have immunoglobulin or complement deposits. There may be variants of MCD (eg, IgM nephropathy or C1q nephropathy). But it is unclear if this represent separate conditions. Moreover, the presence of Ig M deposits in a patient with MCD has been associated with a poorer prognosis and a lower response to glucocorticoids compared to patients with MCD without deposits

Moreover, the proportion of patients with diffuse process effacement in the non-NS group is low (40%) which in my opinion raises questions about whether they had the correct diagnosis.

Answer>

Thank you for your recommendation. 

As you suggested, it was our limitation. We would describe this in discussion as a limitation as below. 

“There are some limitations to this study. This was retrospective study, and the confounders were not strictly controlled. Thus, we only included patients without previous immunosuppressive treatment. In addition, the use of strict inclusion criteria meant that many patients were excluded from the study. As a result, the number of patients included was small. Second, only adults were included in this study, despite MCD being prevalent in children. However, the clinical course of MCD differs in various aspects between adults and children. Thus, it was meaningful to specifically describe the prognosis of MCD in adults. In the same sense, further studies that consider the initial amount of proteinuria are required for patients of all ages. Third, the pathologic findings in some patients were away from classic MCD in our study. Findings were EM findings such as less than 50% of podocyte effacement, and IM findings such as mild deposits. This might because MCD sometimes manifested atypically. Although our study suggested that there was no definite correlation with pathologic severity and the clinical features such as the amount of proteinuria, we still need further findings to prevail this.”(25p)

---

## [Decision Letter · Decision Letter 1]

17 Apr 2023

PONE-D-22-24683R1The outcome of minimal change disease without nephrotic syndromePLOS ONE

Dear Dr. Chin,

Thank you for submitting your manuscript to PLOS ONE. After careful consideration, we feel that it has merit but does not fully meet PLOS ONE’s publication criteria as it currently stands. Therefore, we invite you to submit a revised version of the manuscript that addresses the points raised during the review process. Thank you for submitting a revision addressing comments by reviewers. Please address new comments by reviewer.

Also, consider the following observations:

1. please add how do you identify if the biopsy findings are related to MCD vs. unsampled FSGS? As we are not able to rule out the possibility of unsampled FSGS, a comment on this limitation should be addressed.

2. Clarify in the manuscript that patients in the non-NS group did not receive any other immunosuppression therapy other than the 8 patients who received steroids.

3. For the patients that received other immunosuppressive therapy in the NS group, please clarify the immunosuppression used- specifically, did any of them got rituximab?

4. What is the authors' input in the possibility that some of the subjects in the non-NS group may be just at a different period of the spectrum of MCD, and they may progress to develop NS afterwards?

5. Clarify how many achieved spontaneous remission in both groups, and the period of time in which patients achieved spontaneous remission after diagnosis.

We look forward to receiving your revised manuscript.

Kind regards,

Maria Lourdes Gonzalez Suarez, MD, PhD

Academic Editor

PLOS ONE

Journal Requirements:

Reviewers' comments:

Reviewer's Responses to Questions

**Comments to the Author**

1. If the authors have adequately addressed your comments raised in a previous round of review and you feel that this manuscript is now acceptable for publication, you may indicate that here to bypass the “Comments to the Author” section, enter your conflict of interest statement in the “Confidential to Editor” section, and submit your "Accept" recommendation.

Reviewer #1: All comments have been addressed

2. Is the manuscript technically sound, and do the data support the conclusions?

Reviewer #1: Yes

3. Has the statistical analysis been performed appropriately and rigorously? 

Reviewer #1: Yes

4. Have the authors made all data underlying the findings in their manuscript fully available?

Reviewer #1: Yes

5. Is the manuscript presented in an intelligible fashion and written in standard English?

Reviewer #1: No

6. Review Comments to the Author

Reviewer #1: There are many grammatical errors in the manuscript. These are only some of them. Please have a professional perform a revision of the manuscript.

Lines 102 and 105: Medications instead of medication.

Line 106: "Including serum levels.. delete 'the".

Line 111: According to KDIGO criteria. Would delete "for creatinine measurement".

Line 117; "..in a spot urine sample' not in spot urine.

Line 137-138: ...Findings on EM sounds better, would delete "THE" findings "BY" electron Microscopy.

Line 138- Delete Descripted, use "described as follows" instead.

Line 144- Decrease in eGFR instead "OF"

Line 148... "Divided by the number of years of follow-up" (I believe is what you mean?)

Line 148- We compared the incidences of CR from or relapse of proteinuria, (This sentence isn't clear)

Line 217: "6months"..should be "6 months"

Line 240 .."experienced an increase to UPCR 0.30–0.29 g/g creatinine' change to "..increase in UPCR to 0.30–0.29 g/g creatinine'

Line 250: "was diffused". change to "was diffuse"

Line 252: "and AKI, developed in 2 of them, was relieved". Would use: "and AKI that 2 of them developed, improved.

Line 258- " The days to the first remission were" would use: "The days to first remission"

Line 259- "Except 9 patients" change to "Except for 9 patients"

Line 264- "in non-NS group, when was 1.59..." would use " in non-NS group, and 1.59....."

Line 267: Do not capitalize "The"

7. PLOS authors have the option to publish the peer review history of their article (what does this mean?). If published, this will include your full peer review and any attached files.

Reviewer #1: **Yes: **Aldo Torres-Ortiz

---

## [Author Response · Author response to Decision Letter 1]

26 May 2023

1. please add how do you identify if the biopsy findings are related to MCD vs. unsampled FSGS? As we are not able to rule out the possibility of unsampled FSGS, a comment on this limitation should be addressed.

Thank you for comments. The samples included in this study showed more than 10 glomeruli to maintain sampling adequacy. To Differentiate between MCD and unsampled FSGS, tubulointerstitial damage would have been great to be reported. So, we added this limitation to discussion section. 

(Reference.

Rosenberg AZ, Kopp JB. Focal Segmental Glomerulosclerosis [published correction appears in Clin J Am Soc Nephrol. 2018 Dec 7;13(12):1889]. Clin J Am Soc Nephrol. 2017;12(3):502-517. Doi:10.2215/CJN.05960616)

So, we added some sentences as follows. 

“Third, the pathologic findings in some patients were away from classic MCD in our study. Findings were EM findings such as less than 50% of podocyte effacement, and IM findings such as mild deposits. This might because MCD sometimes manifested atypically. Although our study suggested that there was no definite correlation with pathologic severity and the clinical features such as the amount of proteinuria, we still need further findings to prevail this. “



“Third, the pathologic findings in some patients were away from classic MCD in our study. Findings were EM findings, such as < 50% of podocyte effacement, and IM findings, such as mild deposits. Tubulointerstitial damage to differentiate between MCD and unsampled FSGS was not thoroughly investigated. This might be because MCD is sometimes manifested atypically. However, we tried sampling adequate glomeruli to maintain sampling adequacy. Our study suggested no definite correlation between pathologic severity and clinical features such as the amount of proteinuria; however, we still need further findings to prevail this. “(21p)

2. Clarify in the manuscript that patients in the non-NS group did not receive any other immunosuppression therapy other than the 8 patients who received steroids.

Thank you for recommendation. Sentence about your comment was revised as follows. 

“The 8 patients received steroid treatment.”

“These eight patients received steroid treatment without other immunosuppression therapy.” (13p)

3. For the patients that received other immunosuppressive therapy in the NS group, please clarify the immunosuppression used- specifically, did any of them got rituximab?

Thank you for your comments. Patients in our study did not use rituximab in study period. Other immunosuppressive therapies such as calcineurin inhibitors and cyclosporine were used in some patients. I descripted more details as below. 

“Steroids were administered to 93.2% of patients (55/59) in the NS group. The proteinuria spontaneously improved to a UPCR <0.3 g/g creatine in 4 patients. Among these, 2 patients remained in remission, 1 experienced an increase to UPCR 0.30–0.29 g/g creatinine, and 1 experienced relapse of proteinuria. The maximum dose of prednisolone administered to the 55 patients in NS group was 0.88 ± 0.17 (range: 0.4–1.2) mg/kg/day. Remission of proteinuria was achieved in 51 patients. Eventually, 55 out of 59 patients in NS group achieved a first remission of proteinuria, 4 without treatment and 51 with steroid treatment. Of these, 25 patients ultimately experienced a relapse.”

” Steroids were administered to 93.2% of patients (55/59) in the NS group. The maximum dose of prednisolone administered to the 55 patients in the NS group was 0.88 ± 0.17 (range: 0.4–1.2) mg/kg/day. The proteinuria spontaneously improved to a UPCR <0.3 g/g creatine in four patients. Among these, two patients remained in remission; one increased in UPCR to 0.30-0.29 g/g creatinine, and one experienced a relapse of proteinuria. Thirty-seven out of 55 patients received steroids only, while 14 patients received steroids and calcineurin inhibitors, and four received steroids and cyclophosphamide. Among them, remission of proteinuria was achieved in 51 patients. Eventually, 55 out of 59 patients in the NS group achieved the first remission of proteinuria. Four (6.8%) achieved the first remission without any immunosuppressive treatment. Among 51 patients who used steroids, 34 patients (61.8%) used steroids only, while 13 patients (23.6%) with calcineurin inhibitors, and four patients (7.3%) with cyclophosphamide.” (13p)

4. What is the authors' input in the possibility that some of the subjects in the non-NS group may be just at a different period of the spectrum of MCD, and they may progress to develop NS afterwards?

I sincerely appreciate for your recommendation. This paper was authored with the intention of presenting data that suggests the possibility of progression from a subclinical state to nephrotic syndrome, thereby highlighting the potential for manifestation even in cases that do not initially exhibit clinical symptoms. 

5. Clarify how many achieved spontaneous remission in both groups, and the period of time in which patients achieved spontaneous remission after diagnosis.

16 out of 20 patients (80%) in non-NS group achieved the first CR spontaneously, when 4 out of 59 patients (6.8%) in NS-group did. All 4 patients who had a remission spontaneously were belong to the NS-group. The average period of the first relapse to remission was 33.6 months (±64.31 months). 1 patient among these showed relatively long duration from biopsy day to the day of remission, which was 130.0 months. The amount of proteinuria in this patient decreased, not enough to achieve remission for a while. Except this patient, other three patients showed average 1.42 months (±2.24 months). Meanwhile, others who experienced remission in non-NS group received steroid only.

---

## [Editor Report · Decision Letter 2]

6 Jun 2023

PONE-D-22-24683R2The outcome of minimal change disease without nephrotic syndromePLOS ONE

Dear Dr. Chin,

Thank you for submitting your manuscript to PLOS ONE. After careful consideration, we feel that it has merit but does not fully meet PLOS ONE’s publication criteria as it currently stands. Therefore, we invite you to submit a revised version of the manuscript that addresses the points raised during the review process.

Please correct objectives and conclusions of your study as detailed below.Current conclusion is missing to discuss the initial objectives stated in the introduction. Once the conclusion is editing, please provide this changes in your abstract as well. 

We look forward to receiving your revised manuscript.

Kind regards,

Maria Lourdes Gonzalez Suarez, MD, PhD

Academic Editor

PLOS ONE

Journal Requirements:

Additional Editor Comments:

Thank you for responding to our comments.

There are some other things that would need clarification and editing:

1. In your last sentence of introduction, where you are expressing the objective of your study, you mention: "we compared the clinicopathologic features and outcomes between MCD patients with and without heavy proteinuria for approximately 5 years." However, in your conclusion paragraph your wording is different than your objective: "In conclusion, there was no difference in the prognosis of MCD between patients with and without spontaneous remission. Patients who experience spontaneous remission of MCD should be monitored as closely as those with nephrotic MCD without spontaneous remission." Please unify your wording to match between your objective and your conclusion. Because the current conclusion does not match the objective of the study.

---

## [Author Response · Author response to Decision Letter 2]

11 Jun 2023

Thank you for comments. The sentences in conclusion were corrected into more concise sentences to deliver the author’s point as below.

“In conclusion, there was no overall clinicopathological difference in the prognosis of MCD between patients with and without heavy proteinuria. Therefore, patients who show non-nephrotic proteinuria as an initial manifestation should be monitored as closely as those with nephrotic MCD without spontaneous remission.” (23p)

---

## [Editor Report · Decision Letter 3]

14 Jun 2023

PONE-D-22-24683R3The outcome of minimal change disease without nephrotic syndromePLOS ONE

Dear Dr. Chin,

Thank you for submitting your manuscript to PLOS ONE. After careful consideration, we feel that it has merit but does not fully meet PLOS ONE’s publication criteria as it currently stands. Therefore, we invite you to submit a revised version of the manuscript that addresses the points raised during the review process.

Thank you for addressing our comments.

I have read your changes, and I believe this manuscript has more coherence and clarity. Thank you.

I have some final edit suggestions for your manuscript, please consider:

1. Consider switching the Title to "Outcomes of Minimal Change Disease without Nephrotic Range Proteinuria", which is more representative of the manuscript goals. I hope authors agree.

2. The reference 6 provided by the authors states that all the asymptomatic subjects either had proteinuria of 2g or low serum albumin of 2 (this last one fulfills criteria for NS) but authors describe it in their manuscript as MCD with no NS which is not true, please edit line 63.

3. In line 134, please clarify if all the subjects had EM performed, or what percentage of the study population had EM performed. Consider that if not all subjects had EM, then the diagnosis of MCD is not possible.

Thank you

We look forward to receiving your revised manuscript.

Kind regards,

Maria Lourdes Gonzalez Suarez, MD, PhD

Academic Editor

PLOS ONE

Journal Requirements:

Additional Editor Comments:

Thank you for addressing my comments.

I have read your changes, and I believe this manuscript has more coherence and clarity. Thank you.

I have some final edit suggestions for your manuscript, please consider:

1. Consider switching the Title to "Outcomes of Minimal Change Disease without Nephrotic Range Proteinuria", which is more representative of the manuscript goals. I hope authors agree.

2. The reference 6 provided by the authors states that all the asymptomatic subjects either had proteinuria of 2g or low serum albumin of 2 (this last one fulfills criteria for NS) but authors describe it in their manuscript as MCD with no NS which is not true, please edit line 63.

3. In line 134, please clarify if all the subjects had EM performed, or what percentage of the study population had EM performed. Consider that if not all subjects had EM, then the diagnosis of MCD is not possible.

Thank you

---

## [Author Response · Author response to Decision Letter 3]

22 Jul 2023

1. Consider switching the Title to "Outcomes of Minimal Change Disease without Nephrotic Range Proteinuria", which is more representative of the manuscript goals. I hope authors agree.

Thank you for the editors’ recommendation. We agree to change the title as you suggested. Short title was also changed to “Minimal Change Disease with or without Nephrotic Range Proteinuria”. Thank you.

2. The reference 6 provided by the authors states that all the asymptomatic subjects either had proteinuria of 2g or low serum albumin of 2 (this last one fulfills criteria for NS) but authors describe it in their manuscript as MCD with no NS which is not true, please edit line 63.

I sincerely thank you for your recommendation. After reviewing the reference, we found that it was not appropriate to deliver our opinion in introduction due to misinterpretation. We changed the sentence as follows. Reference numbers were changed according to the deletion of the reference in Introduction section. 

“In adults, MCD exhibits a relatively large level of proteinuria compared with other primary glomerulonephritis. However, the long-term prognosis of adult MCD without nephrotic syndrome as an initial manifestation is unclear. There are some studies on the prognosis of MCD without nephrotic syndrome in children [6]. Patients with an initial presentation of proteinuria < 2 g/day/m2 did not experience relapses within 1 year, while those with nephrotic syndrome did. Children with lower levels of proteinuria were also more sensitive to steroid treatment. However, the prognosis of MCD with typical pathologic findings needs to be defined in terms of clinical characteristics, response rates, and relapse rates in patients with and without nephrotic syndrome. Therefore, we compared the clinicopathologic features and outcomes between MCD patients with and without heavy proteinuria for approximately 5 years.”

“In adults, MCD exhibits a relatively large level of proteinuria compared with other primary glomerulonephritis. Until this study, most studies have focused on heavy proteinuria diagnosed with MCD. Therefore, the long-term prognosis of adult MCD without nephrotic syndrome as an initial manifestation is unclear. To fully understand MCD, the prognosis of MCD with typical pathologic findings needs to be defined in terms of clinical characteristics, response rates, and relapse rates in patients with and without nephrotic syndrome. In this study, we compared the clinicopathologic features and outcomes between MCD patients with and without heavy proteinuria for approximately 5 years.” (line 61–67, 4p) 

3. In line 134, please clarify if all the subjects had EM performed, or what percentage of the study population had EM performed. Consider that if not all subjects had EM, then the diagnosis of MCD is not possible.

Thank you for recommendation. We added the sentence as follows. 

Findings on EM were described as follows; the presence of electron-dense deposits in the area of the mesangium, subendothelium, and subepithelium; and severity of foot process effacement of the podocytes, which were reported as none, focal (mild, moderate, moderate to severe, severe), or diffuse. 

“After EM analysis was done in all biopsy samples, findings on EM were described as follows; the presence of electron-dense deposits in the area of the mesangium, subendothelium, and subepithelium; and severity of foot process effacement of the podocytes, which were reported as none, focal (mild, moderate, moderate to severe, severe), or diffuse.” (line 131–135, 8p)

---

## [Editor Report · Decision Letter 4]

28 Jul 2023

Outcomes of minimal change disease without nephrotic range proteinuria

PONE-D-22-24683R4

Dear Dr. Chin,

We’re pleased to inform you that your manuscript has been judged scientifically suitable for publication and will be formally accepted for publication once it meets all outstanding technical requirements.

Kind regards,

Maria Lourdes Gonzalez Suarez, MD, PhD

Academic Editor

PLOS ONE

Additional Editor Comments (optional):

Thank you for making the changes to the manuscript. I appreciate your patience. I agree with your edits and comments.

---

## [Editor Report · Acceptance letter]

10 Aug 2023

PONE-D-22-24683R4 

Outcomes of minimal change disease without nephrotic range proteinuria 

Dear Dr. Chin:

I'm pleased to inform you that your manuscript has been deemed suitable for publication in PLOS ONE. Congratulations! Your manuscript is now with our production department. 

Kind regards, 

on behalf of

Dr. Maria Lourdes Gonzalez Suarez 

Academic Editor

PLOS ONE